# OpenReview forum: "Leveraging the Context through Multi-Round Interactions for Jailbreaking Attacks"
_ICLR.cc/2025/Conference — ICLR 2025 Conference Withdrawn Submission_

### Official Review · Reviewer_ucTn · 2024-11-02

**Soundness:** 2
**Presentation:** 2
**Contribution:** 2
**Rating:** 3
**Confidence:** 5

**Summary:**

This paper presents a multi-turn approach to interact with large language models (LLMs) and elicit a jailbreak condition by using an auxiliary LLM to automatically generate a sequence of benign prompts. The authors validate their proposed method on both closed and open-source LLMs, reporting jailbreak success rates of ≥ 90% on two open-source LLMs and ≥ 80% on two closed-source LLMs. Additionally, the paper explores cases where the proposed method does not achieve high jailbreak percentages on one closed-source and one open-source LLM. The approach aims to automate the creation of the attack prompt sequence, contributing to the automation of jailbreak attacks on LLMs.

**Strengths:**

- The paper is well-structured and thoroughly validates the proposed method on multiple LLMs, providing a comprehensive assessment of its effectiveness.
- The authors conduct an extensive empirical analysis, highlighting the jailbreak potential across different LLMs and presenting their findings in a detailed and systematic manner.

**Weaknesses:**

- The concept of multi-turn attacks on LLMs is already well-established in the literature (e.g., [1, 2, 3, 4]), with [2] particularly exploring the use of context to craft jailbreak sequences. Authors should clarify how this attack is different from the contextual attacks in [2]
- The method for generating benign prompts closely resembles that in [5], where in-context learning capabilities of the LLM are utilized to generate new attack prompts based on given examples. Authors should clearly state the key difference in their approach for generating prompts compared to [5]
- Section 4.1, which discusses evaluation metrics, mentions that the judge function relies entirely on human evaluation. However, there is insufficient clarity regarding the number of examples used to compute the jailbreak percentage and how human evaluation scores correlate with an automated LLM judge score. Authors should clearly specify the following points in regard to the paper:
1. The exact number of examples used to compute jailbreak percentages
2. Details on how many human evaluators were involved
3. Any analysis comparing human evaluation scores to automated LLM judge scores, if available

References:
1. [Multi-turn and contextual jailbreak attacks research](https://arxiv.org/abs/2408.15221)
2. [CFA: Contextual attacks on LLMs](https://arxiv.org/abs/2408.04686)
3. [RedQueen: Multi-turn attacks](https://arxiv.org/pdf/2409.17458)
4. [MART: Adversarial prompt sequences](https://arxiv.org/pdf/2311.07689)
5. [PAIR: In-context attack prompt generation](https://arxiv.org/abs/2310.08419)
6. [Prompt-Guard-86M](https://huggingface.co/meta-llama/Prompt-Guard-86M)
7. [Llama-Guard-3-8B](https://huggingface.co/meta-llama/Llama-Guard-3-8B)

**Questions:**

Questions and Suggestions
1. Clarification on Sequential Tailoring: Section 3.2 claims that the model’s responses to preliminary questions are used to tailor subsequent prompts for attack purposes. However, further details on how these responses influence the attack's progression would add clarity to the method and demonstrate the practical application of this approach.

2. Details on Evaluation and Human Judgement: The number of examples used to compute jailbreak percentages is not specified, making it difficult to assess the reliability of the reported success rates. Additionally, incorporating an automated LLM-based judge to compare human evaluation scores could enhance the objectivity of the results and add depth to the validation process.

3. Further Validation Against Advanced Detectors: The current evaluation does not include finetuned adversarial prompt detectors, which could provide a more comprehensive test of the method’s robustness. Testing against finetuned models, such as [6] and [7], specifically designed to detect malicious prompts, would offer valuable insights into the effectiveness of the proposed attack.

References:
1. [Multi-turn and contextual jailbreak attacks research](https://arxiv.org/abs/2408.15221)
2. [CFA: Contextual attacks on LLMs](https://arxiv.org/abs/2408.04686)
3. [RedQueen: Multi-turn attacks](https://arxiv.org/pdf/2409.17458)
4. [MART: Adversarial prompt sequences](https://arxiv.org/pdf/2311.07689)
5. [PAIR: In-context attack prompt generation](https://arxiv.org/abs/2310.08419)
6. [Prompt-Guard-86M](https://huggingface.co/meta-llama/Prompt-Guard-86M)
7. [Llama-Guard-3-8B](https://huggingface.co/meta-llama/Llama-Guard-3-8B)

**Details Of Ethics Concerns:**

Not Needed

---

### Official Review · Reviewer_n3UT · 2024-11-02

**Soundness:** 3
**Presentation:** 3
**Contribution:** 1
**Rating:** 3
**Confidence:** 5

**Summary:**

This paper proposes a multi-turn jailbreak, which crafts a series of harmless adversarial queries to guide the victim model to output harmful content. Experimental results demonstrate the effectiveness of their attack method against different large language models.

**Strengths:**

+ The presentation of the methodology part is clear and easy to follow.
+ The experimental results are comprehensive, which include both advanced closed-source and open-source LLMs.

**Weaknesses:**

- **Lack of Novelty**: The core idea of this paper closely resembles that of Crescendo [1], a multi-turn jailbreak method that starts with an innocuous query about a harmful task and gradually guides the model to generate harmful content through benign incremental steps. Given this similarity, the concept presented in this paper does not appear to introduce significant novelty over Crescendo.

   Furthermore, the implementation in this paper also mirrors Crescendo’s approach. Crescendo employs in-context learning with a series of carefully crafted examples—a strategy similarly adopted in this paper. For instance, Crescendo’s attack sequence begins with “introduce the history of [harmful target]” (Figure 1), while this paper uses a similar prompt, “Write a paper on the history of [harmful target],” as the starting point of its attack (Figure 1).

   It would be valuable for the authors to elaborate on how their approach diverges from or improves upon Crescendo's methodology. Are there any distinctive methodological or experimental results that could underscore the novelty of this work?

- **Insufficient Experiments**: The experiments in this paper are limited to a subset of the AdvBench benchmark, while other benchmarks, such as HarmBench [2] and SorryBench [3], offer more balanced and less noisy alternatives. Including results from HarmBench or SorryBench would enhance the robustness and comprehensiveness of the experimental evaluation, and the authors should consider incorporating these benchmarks in future work or as additional experiments.

- **Lack of Comparisons with Other Multi-Turn Attacks**: Besides Crescendo, several other multi-turn attack methods have recently emerged, including [4, 5, 6]. This paper would benefit from a concise comparison with these approaches, outlining key similarities or differences, and including a discussion of these methods in the related work section to provide a more comprehensive review of the landscape in multi-turn attacks.

1. *Great, Now Write an Article About That: The Crescendo Multi-Turn LLM Jailbreak Attack.*
2. *HarmBench: A Standardized Evaluation Framework for Automated Red Teaming and Robust Refusal.*
3. *SORRY-Bench: Systematically Evaluating Large Language Model Safety Refusal Behaviors.*
4. *Chain of Attack: A Semantic-Driven Contextual Multi-Turn Attacker for LLMs.*
5. *Speak Out of Turn: Safety Vulnerability of Large Language Models in Multi-Turn Dialogue.*
6. *Imposter.ai: Adversarial Attacks with Hidden Intentions Towards Aligned Large Language Models.*

**Questions:**

1. What is the human judge setup?
2. Why the attack success rate of Contextual Interaction Attack against Claude-2 is much lower than other models? Are there any distinctive behaviors of Claude-2 compared with other models?

I would like to raise my score if the authors could address my concerns and questions.

---

### Official Review · Reviewer_1sbm · 2024-11-02

**Soundness:** 3
**Presentation:** 3
**Contribution:** 4
**Rating:** 6
**Confidence:** 4

**Summary:**

This paper proposes a novel multi-turn attack - Contextual Interaction Attack, using the model interactions (here is the benign preliminary questions to interact with the LLM) to elicit harmful responses. Specifically, this multi-turn attack leverages the autoregressive nature of LLMs to guide the model construct a context that is semantically aligned with the attack target, based on the question-responses pair.

**Strengths:**

1. Interesting topic and insightful research, especially the mechanism analysis
2. Well-written and visual logic
3. Abundant and convincing experiment

**Weaknesses:**

1. Lack of some detailed evidence to support the demonstration
2. Lack of comparison with the prevalent multi-turn attack Crescendo

**Questions:**

1. Explanation: Please give an illustrative example to support your demonstration - ” compared to harmful information provided by the user (exogenous input), LLMs tend to perceive their own generated content (endogenous output) as safer.”

2. Experiment: Please add Crescendo[a] as a multi-turn attack baseline for comparison in experiment section

[a] Russinovich, Mark, Ahmed Salem, and Ronen Eldan. "Great, now write an article about that: The crescendo multi-turn llm jailbreak attack." arXiv preprint arXiv:2404.01833 (2024).

---

### Official Review · Reviewer_Cffr · 2024-11-03

**Soundness:** 2
**Presentation:** 2
**Contribution:** 2
**Rating:** 3
**Confidence:** 3

**Summary:**

This paper presents the Contextual Interaction Attack (CIA) against LLMs, which involves prepending a series of prompt, response pairs that gradually lead to a harmful question. The prompts in these conversations are generated via few-shot prompting the Wizard-70b LLM, and response generated by the target model. The authors show results competitive with other attacks on 3 open-source and 3 closed-source LLMs.

**Strengths:**

The topic of this paper is important, jailbreaking attacks demonstrate LLMs are not robust and easily give harmful content outside of a single-turn question.

The work is relatively easy to understand and the authors provide a good amount of ablation studies to justify the design of their methods. It takes inspiration from prior work to leverage the context, but does not use any optimization against a specific model.

**Weaknesses:**

Novelty:
The main results on 50 prompts do not show significantly more success than prior attacks. The open source models evaluated are not exciting, they’re somewhat outdated and not claimed to be robust to attacks. This work would benefit from attacks on more recent models such as Llama 3 or 3.1, or attacking a larger model/models with refusal training.
The method is fairly simple, it boils down to prompting another LLM to mimic how a human jailbreaker would attack an LLM. It would be great to elaborate more on the novelty of this approach relative to recent work on multi-turn jailbreaking.

Soundness:
A human judge was used on AdvBench without much detail given, this affects the soundness of the paper. Please elaborate more. Have you considered testing your method with a controlled judge, such as HarmBench [1], which has a trained static classifier?
The transferability experiments (Table 2) feel contrived. (1) Attacks such as PAIR don’t assume a threat model to transfer the instance-level outputs of the attack, it is not a fair comparison. However, the attack itself only needs black box access, it doesn’t need to transfer the instance-level output. (2) The closed-source models tested have specific defenses implemented against popular attacks such as GCG, it is not a fair comparison to measure transferability. The authors should report the results in the original attack paper whenever possible, which shows a much stronger transferability attack against GPT-3.5 and GPT-4 models.

Defenses:
Perplexity based defenses tested are specifically designed against GCG-style attacks, it is expected that they are not actually detecting harm. This work would benefit from showing strong results on general defenses such as DERTA [2] or Llama Guard [3], or more robustly trained models such as CYGNET.

Presentation:
Page 1 line 45: “jailbreak prompts generated by automated attacks
often do not perform well when transferred to other LLMs, which is a major bottleneck”. This contradicts the citation on GCG immediately prior, which is transferable to other LLMs. The reason why GCG performs worse in this paper when transferred is likely due to model providers providing specific defenses against it. This does automated attacks transfer poorly.
Related work section only cites 3 papers involving jailbreaking LLMs, the rest are on background information such as in-context learning and autoregressive LMs that should be assumed as basic knowledge among this community. It would be good to further contextualize the novelty of this attack relative to prior work.
Minor: Figure 1 has a typo “Repharse” -> “Rephrase”.

[1] Mazeika, M., Phan, L., Yin, X., Zou, A., Wang, Z., Mu, N., ... & Hendrycks, D. (2024). Harmbench: A standardized evaluation framework for automated red teaming and robust refusal. arXiv preprint arXiv:2402.04249.
[2] Yuan, Y., Jiao, W., Wang, W., Huang, J. T., Xu, J., Liang, T., ... & Tu, Z. (2024). Refuse whenever you feel unsafe: Improving safety in llms via decoupled refusal training. arXiv preprint arXiv:2407.09121.
[3] Inan, H., Upasani, K., Chi, J., Rungta, R., Iyer, K., Mao, Y., ... & Khabsa, M. (2023). Llama guard: Llm-based input-output safeguard for human-ai conversations. arXiv preprint arXiv:2312.06674.

**Questions:**

Wizard-70b was used as the auxiliary model. This is a strange setup when attacking weaker models such as Vicuna and LLaMA-2-7b. Why would you need to attack a weaker model when you already have a stronger model that is capable of manipulating the weaker model?
Clarify the process for training the human judge to classify harm. Is it just a single human? Is the judge independent of the authors with a single or double-blind setup?

---

### Note · Authors · 2024-11-15

I have read and agree with the venue's withdrawal policy on behalf of myself and my co-authors.